# Why Deep Models Often Cannot Beat Non-deep Counterparts on Molecular Property Prediction?

Jun Xia [* 1]  Lecheng Zhang [* 1]  Xiao Zhu [* 1]  Stan Z. Li [1]

## Abstract

Molecular property prediction (MPP) is a crucial task in the drug discovery pipeline, which has recently gained considerable attention thanks to advances in deep neural networks. However, recent research has revealed that deep models struggle to beat traditional non-deep ones on MPP. In this study, we benchmark 12 representative models (3 non-deep models and 9 deep models) on 14 molecule datasets. Through the most comprehensive study to date, we make the following key observations: **(i)** Deep models are generally unable to outperform non-deep ones; **(ii)** The failure of deep models on MPP cannot be solely attributed to the small size of molecular datasets. What matters is the irregular molecule data pattern; **(iii)** In particular, tree models using molecular fingerprints as inputs tend to perform better than other competitors. Furthermore, we conduct extensive empirical investigations into the unique patterns of molecule data and inductive biases of various models underlying these phenomena.

## 1. Introduction

Molecular Property Prediction (MPP) is a critical task in drug discovery, aimed at identifying molecules with desirable pharmacological and ADMET (absorption, distribution, metabolism, excretion, and toxicity) properties. Machine learning models have been widely used in this fast-growing field, with two types of models being commonly employed: traditional non-deep models and deep models. In non-deep models, molecules are fed into traditional machine learning models such as Random Forest and Support Vector Machine in the format of computed or handcrafted molecular fingerprints (Todeschini & Consonni, 2010). The other group utilizes deep models to extract expressive representations for molecules in a data-driven manner. Specifically, the Multi-Layer Perceptron (MLP) could be applied to computed or handcrafted molecular fingerprints; Sequence-based neural architectures including Recurrent Neural Networks (RNNs) (Medsker & Jain, 1999), 1D Convolutional Neural Networks (1D CNNs) (Gu et al., 2018), and Transformers (Honda et al., 2019; Rong et al., 2020) are exploited to encode molecules represented in Simplified Molecular-Input Line-Entry System (SMILES) strings (Weininger et al., 1989). Later, it is argued that molecules can be naturally represented in graph structures with atoms as nodes and bonds as edges. This inspires a line of works to leverage such structured inductive bias for better molecular representations (Gilmer et al., 2017; Xiong et al., 2019; Yang et al., 2019; Song et al., 2020). The key advancements underneath these approaches are Graph Neural Networks (GNNs), which consider graph structures and attributive features simultaneously by recursively aggregating node features from neighborhoods (Kipf & Welling, 2017; Velickovic et al., 2018; Hamilton et al., 2017). More recently, researchers incorporate 3D conformations of molecules into their representations for better performance, whereas pragmatic considerations such as calculation cost, alignment invariance, and uncertainty in conformation generation limited the practical applicability of these models (Axen et al., 2017; Gasteiger et al., 2020; Schuett et al., 2017; Gasteiger et al., 2021; Liu et al., 2022). We summarize the widely-used molecular descriptors and their corresponding models in our benchmark, as shown in Figure 1. Despite the fruitful progress, previous studies (Mayr et al., 2018; Yang et al., 2019; Valsecchi et al., 2022; Jiang et al., 2021; van Tilborg et al., 2022; Janela & Bajorath, 2022) have observed that deep models struggled to outperform non-deep ones on molecules. However, these studies neither consider the emerging powerful deep models (e.g., Transformer (Honda et al., 2019), SphereNet (Liu et al., 2021)) nor explore various molecular descriptors (e.g., 3D molecular graph). Also, they did not investigate the reasons why deep models often fail on molecules.

To narrow this gap, we present the most comprehensive benchmark study on molecular property prediction to date, with a precise methodology for dataset inclusion and hyperparameter tuning. Our empirical results confirm the

---

*Equal contribution [1]Westlake University, Hangzhou, China. Correspondence to: Jun Xia <xiajun@westlake.edu.cn>, Stan Z. Li <stan.zq.li@westlake.edu.cn>.

*Workshop on Interpretable ML in Healthcare at International Conference on Machine Learning (ICML)*, Honolulu, Hawaii, USA. 2023. Copyright 2023 by the author(s).

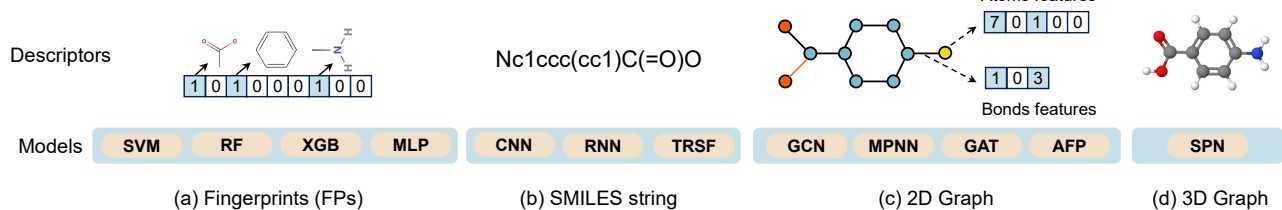

*Figure 1.* Exemplary molecular descriptors and their corresponding models in our benchmark. **SVM**: Support Vector Machine (Zernov et al., 2003); **RF**: Random Forest (Svetnik et al., 2003); **XGB**: eXtreme Gradient Boosting (Chen & Guestrin, 2016); **MLP**: Multi-Layer Perceptron; **CNN**: 1D Convolution Neural Network (Kimber et al., 2021); **RNN**: Recurrent Neural Network (GRU) (Mulder et al., 2015); **TRSF**: TRanSFormer (Vaswani et al., 2017); **GCN**: Graph Convolution Network (Kipf & Welling, 2017); **MPNN**: Message-Passing Neural Network (Gilmer et al., 2017); **GAT**: Graph Attention neTwork (Velickovic et al., 2018); **AFP**: Attentive FP (Xiong et al., 2020); **SPN**: SPhereNet (Liu et al., 2022). The above-mentioned abbreviations are applicable throughout the entire paper.

observations of previous studies, namely that deep models generally cannot outperform traditional non-deep counterparts. Moreover, we observe several interesting phenomena that challenge the prevailing beliefs of the community, which can guide optimal methodology design for future studies. Furthermore, we transform the original molecular data to observe the performance changes of various models, uncovering the unique patterns of molecular data and the differing inductive biases of various models. These in-depth empirical studies shed light on the benchmarking results.

## 2. Benchmarking Results.

In this section, we present a benchmark on 14 molecular datasets with 12 representative models.

### 2.1. Observations

Table 1 documents the benchmark results for various models and datasets, from which we can make the following *Observations*:

*Observation 1.* **Deep models underperform non-deep counterparts in most cases.**
As can be observed in Table 1, non-deep models rank as the top one on 10/14 datasets. On some datasets such as MUV, QM7, and BACE, three non-deep models can even beat any deep models.

*Observation 2.* **It is irregular data patterns, NOT solely the small size of molecular datasets to blame for the failure of deep models!**
Intuitively, many previous works (Goh et al., 2017; Yang et al., 2019) pointed out that the small size of molecular datasets could be a bottleneck for deep learning models. Here, we provide a second voice to such pre-dominant beliefs with empirical evidence. As shown in Table 1, all the non-deep models can outperform any deep ones on some larger-scale datasets (e.g., MUV and QM 7). However, in some small datasets (e.g., ClinTox and ESOL), some deep models can beat partial non-deep ones. Therefore, what

matters is the irregular molecule data pattern, not solely the dataset size. We will provide an in-depth analysis to the unique molecule data pattern in Sec. 3.

*Observation 3.* **Tree models (XGB and RF) exhibit a particular advantage over other models.**
In the experiments shown in Table 1, we can see that the tree-based models consistently rank among the top three on each dataset. Additionally, tree models rank as the top one on 8/15 datasets. We will explore why tree models are well-suited for molecular fingerprints in Sec. 3.

## 3. Why above phenomena would occur?

In this section, we attempt to understand which characteristics of molecular data lead to the failure of powerful deep models. Also, we aim to understand the inductive biases of tree models that make them well-suited for molecules, and how they differ from the inductive biases of deep models.

*Explanation 1.* **Unlike image data, molecular data patterns are non-smooth. Deep models struggle to learn non-smooth target functions that map molecules to properties.**
We design two experiments to verify the above explanation, i.e., increasing or decreasing the level of data smoothing in the molecular datasets. Firstly, we transform the molecular data by smoothing the labels based on similarities between molecules. Specifically, let $\mathcal{D}$ denote the molecular dataset and $(x_i, y_i) \in \mathcal{D}$ be $i$-th molecule and its label, we smooth the target function as follows,

$$\widehat{y_i} = \frac{\sum_{x_j \in \mathcal{N}_{x_i}} s(x_i, x_j) y_j}{\sum_{x_j \in \mathcal{N}_{x_i}} s(x_i, x_j)}, \tag{1}$$

where $s(\cdot, \cdot)$ denotes the Tanimoto coefficient of the extended connectivity fingerprints (ECFP) between two molecules that can be considered as their structural similarity. $\mathcal{N}_{x_i}$ is the $k$-nearest neighbor set of $x_i$ (including $x_i$) picked from the whole dataset based on the structural similarities. $\widehat{y_i}$ denotes the label after smoothing. We smooth all the molecules in the dataset in this way and use

*Table 1.* The comparison of representative models on multiple molecular datasets. The standard deviations can be seen in the appendix for the limited space. **No.**: Number of the molecules in the datasets. The top-3 performances on each dataset are highlighted with the grey background. The best performance is highlighted with **bold**. Kindly note that **'TRSF'** denotes the transformer that has been pre-trained on 861, 000 molecular SMILES strings. The results on QM 9 can be seen in the appendix.

| Dataset (No.) | Metric | SVM | XGB | RF | CNN | RNN | TRSF | MLP | GCN | MPNN | GAT | AFP | SPN |
|---|---|---|---|---|---|---|---|---|---|---|---|---|---|
| BACE (1,513) | AUC_ROC | 0.886 | **0.896** | 0.890 | 0.815 | 0.559 | 0.835 | 0.887 | 0.880 | 0.846 | 0.886 | 0.879 | 0.882 |
| HIV (40,748) | AUC_ROC | 0.817 | 0.823 | 0.826 | 0.733 | 0.639 | 0.748 | 0.791 | **0.834** | 0.814 | 0.812 | 0.819 | 0.818 |
| BBBP (2,035) | AUC_ROC | 0.913 | **0.926** | 0.923 | 0.760 | 0.693 | 0.897 | 0.918 | 0.915 | 0.872 | 0.902 | 0.893 | 0.905 |
| ClinTox (1,475) | AUC_ROC | 0.879 | 0.919 | 0.933 | 0.685 | 0.813 | **0.963** | 0.890 | 0.889 | 0.868 | 0.891 | 0.907 | 0.912 |
| SIDER (1,366) | AUC_ROC | 0.626 | 0.638 | **0.644** | 0.591 | 0.515 | 0.641 | 0.617 | 0.633 | 0.603 | 0.614 | 0.620 | 0.613 |
| Tox21 (7,811) | AUC_ROC | 0.820 | 0.837 | 0.838 | 0.766 | 0.734 | 0.817 | 0.834 | 0.830 | 0.816 | 0.829 | **0.845** | 0.827 |
| ToxCast (8,539) | AUC_ROC | 0.725 | 0.785 | 0.778 | 0.735 | 0.74 | 0.780 | 0.781 | 0.767 | 0.736 | 0.768 | **0.788** | 0.772 |
| MUV (93,087) | AUC_PRC | **0.093** | 0.072 | 0.069 | 0.045 | 0.094 | 0.059 | 0.018 | 0.056 | 0.019 | 0.055 | 0.044 | 0.058 |
| SARS-CoV-2 (14,332) | AUC_ROC | 0.599 | **0.700** | 0.686 | 0.688 | 0.649 | 0.643 | 0.638 | 0.646 | 0.640 | 0.683 | 0.651 | 0.663 |
| ESOL (1,127) | RMSE | 0.676 | **0.583** | 0.647 | 2.569 | 1.511 | 0.718 | 0.653 | 0.773 | 0.695 | 0.661 | 0.594 | 0.671 |
| Lipop (4,200) | RMSE | 0.683 | **0.585** | 0.626 | 1.016 | 1.207 | 0.947 | 0.633 | 0.665 | 0.669 | 0.680 | 0.664 | 0.630 |
| FreeSolv (639) | RMSE | 1.063 | **0.715** | 1.014 | 2.275 | 2.205 | 1.504 | 1.046 | 1.316 | 1.327 | 1.304 | 1.139 | 1.159 |
| QM7 (6,830) | MAE | **42.814** | 52.726 | 51.403 | 81.165 | 158.160 | 64.363 | 86.060 | 64.530 | 107.013 | 78.217 | 59.973 | 55.727 |
| QM8 (21,786) | MAE | 0.0364 | 0.0126 | **0.0098** | 0.0205 | 0.0295 | 0.0232 | 0.0104 | 0.0154 | 0.0109 | 0.0187 | **0.0098** | 0.0103 |

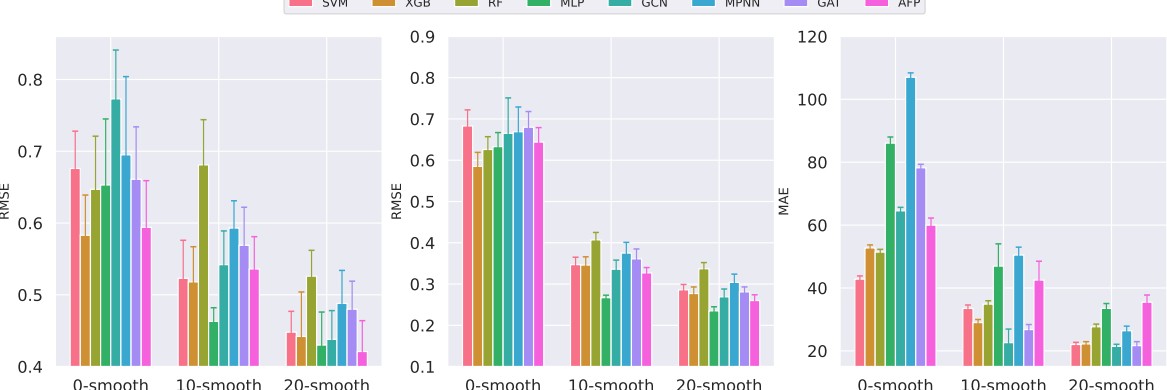

*Figure 2.* The performance of various models on the smoothed datasets. **Left**: ESOL (Regression); **Middle**: Lipop (Regression); **Right**: QM7 (Regression). We only smooth the regression datasets because the labels of classification datasets are not suitable for smoothing.

the smoothed label $\hat{y}_i$ to train the models. The results are shown in Figure 2, where '0-smooth' denotes the original datasets. '10-smooth' and '20-smooth' mean $k = 10$ and $k = 20$, respectively. As can be observed, the performance of deep models improves dramatically as the level of dataset smoothing increases, and many deep models including MLP, GCN, and AFP can even beat non-deep ones after smoothing. These phenomena indicate that deep models are more suitable for the smoothed datasets.

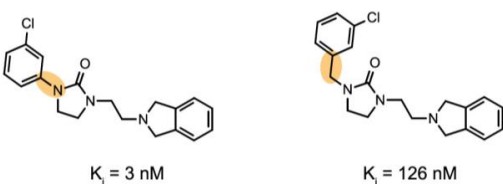

*Figure 3.* Examplary of Activity Cliffs (ACs) on the target named dopamine D3 receptor (D3R). $K_i$ means the bioactivity values. This figure is adapted from Derek van Tilborg's work (van Tilborg et al., 2022).

Secondly, we decrease the level of data smoothing using the concept of activity cliff (Maggiora, 2006; Stumpfe & Bajorath, 2012) from chemistry, which means a situation where small changes in the chemical structure of a drug lead to significant changes in its bioactivity. We provide an example activity cliff pairs in Figure 3. Apparently, the target function of activity cliffs that map molecules to the activity values is less smooth than normal molecular datasets. We then evaluate the models on the activity cliff datasets (van Tilborg et al., 2022). The test set contains molecules that are chemically similar to those in the training set but exhibit either a large difference in bioactivity (cliff molecules) or similar bioactivity (non-cliff molecules). As shown in Table 2, the non-deep models consistently outperform deep ones on these activity cliff datasets. Furthermore, it is worth noting that deep models exhibit a similar level of performance on both non-cliff and cliff molecules, while non-deep models experience significant changes in performance when transitioning from non-cliff to cliff molecules. These phe-

*Table 2.* $\text{RMSE}_{nc}$ and $\text{RMSE}_c$ are the prediction RMSE on non-cliff molecules and cliff molecules, respectively. $\Delta\mathcal{R} = (\text{RMSE}_c - \text{RMSE}_{nc}) / \text{RMSE}_{nc} \times 100\%$. The top-3 performances and the best performance are highlighted with grey background and **bold**.

| Target name (Response type) | Metric | SVM | XGB | RF | CNN | RNN | TRSF | MLP | GCN | MPNN | GAT | AFP |
|---|---|---|---|---|---|---|---|---|---|---|---|---|
| CB1 (Agonism $EC_{50}$) | $\text{RMSE}_{nc}$ | 0.652 | 0.623 | **0.619** | 0.934 | 0.712 | 0.785 | 0.707 | 0.932 | 0.938 | 0.960 | 0.909 |
| | $\text{RMSE}_c$ | 0.773 | **0.767** | 0.770 | 0.944 | 0.823 | 0.888 | 0.807 | 0.992 | 0.989 | 0.975 | 0.967 |
| | $\Delta\mathcal{R}$ | 18.55% | 23.11% | 24.39% | 1.15% | 15.59% | 13.12% | 14.1% | 6.37% | 5.47% | 1.55% | 6.35% |
| DAT (Inhibition $K_i$) | $\text{RMSE}_{nc}$ | 0.589 | 0.579 | **0.577** | 0.871 | 0.692 | 0.801 | 0.664 | 0.927 | 0.820 | 0.995 | 0.865 |
| | $\text{RMSE}_c$ | 0.744 | **0.696** | 0.730 | 0.894 | 0.783 | 0.934 | 0.792 | 1.003 | 0.921 | 1.042 | 0.995 |
| | $\Delta\mathcal{R}$ | 26.30% | 20.18% | 26.64% | 2.48% | 13.15% | 16.70% | 19.40% | 8.23% | 12.38% | 4.74% | 15.11% |
| PPAR$\alpha$ (Agonism $EC_{50}$) | $\text{RMSE}_{nc}$ | **0.535** | 0.552 | 0.561 | 0.854 | 0.696 | 0.799 | 0.606 | 0.856 | 0.833 | 0.892 | 0.749 |
| | $\text{RMSE}_c$ | **0.671** | 0.678 | 0.685 | 0.962 | 0.825 | 0.968 | 0.713 | 0.870 | 0.872 | 0.929 | 0.823 |
| | $\Delta\mathcal{R}$ | 25.42% | 22.83% | 22.10% | 12.69% | 15.64% | 21.26% | 17.77% | 1.72% | 4.78% | 4.21% | 9.90% |
| DOR (Inhibition $K_i$) | $\text{RMSE}_{nc}$ | 0.598 | 0.592 | **0.591** | 0.938 | 0.893 | 0.873 | 0.663 | 1.095 | 0.958 | 1.102 | 1.018 |
| | $\text{RMSE}_c$ | 0.861 | 0.854 | **0.836** | 1.098 | 1.036 | 1.032 | 0.874 | 1.259 | 1.152 | 1.281 | 1.179 |
| | $\Delta\mathcal{R}$ | 43.98% | 44.14% | 41.46% | 17.06% | 16.01 % | 18.26% | 31.85% | 14.93% | 20.27% | 16.26% | 15.83% |

nomena indicate that deep models are less sensitive to subtle structural changes and struggle to learn non-smooth target functions compared with tree models, especially the activity cliff cases. Our explanation is consistent with the conclusions in deep learning theory (Rahaman et al., 2019), i.e., deep models struggle to learn high-frequency components of the target functions. However, tree models can learn piece-wise target functions, and do not exhibit such bias. Our explorations uncover several promising avenues to enhance deep models' performance on molecules: smoothing the target functions or improving deep models' ability to learn the non-smooth target functions.

*Explanation 2.* **Deep models mix different dimensions of molecular features, whereas tree models make decisions based on each dimension of the features separately.** Typically, features in molecular data carry meanings individually. Each dimension of molecular fingerprints often indicates whether a certain substructure is present in the molecule; each dimension of nodes/edges features in molecular graph data indicates a specific characteristic of the atoms/bonds (e.g., atom/bond type, atom degree). To verify the above explanation, we mix the different dimensions of molecular features $x_i \in \mathbb{R}^d$ using an orthogonal transformation before feeding them into various models,

$$\widehat{x_i} = \mathcal{Q}x_i, \tag{2}$$

where $\mathcal{Q} \in \mathbb{R}^{d \times d}$ is the orthogonal matrix and $\widehat{x_i}$ is the molecular feature after transformation. Kindly note that the meaning of $x_i$ depends on the input molecular descriptors in the experiments. Specifically, for SVM, XGB, RF, and MLP, $x_i$ denotes the molecular fingerprints; for GNN models, $x_i$ can denote the atom features and bond features in the molecular graphs, i.e., we apply orthogonal transformations to both the atom features and bond features. As can be observed in Figure 4, the performance of tree models deteriorates dramatically and falls behind most deep models

after the orthogonal transformation. It is because each dimension of $\widehat{x_i}$ is a convex combination of all the dimensions of $x_i$ according to the matrix-vector product rule. In other words, the molecular features after orthogonal transformation no longer carry meanings individually, accounting for the failure of tree models that make decisions based on each dimension of the features separately. The learning style of tree models is more suitable for molecular data because only a handful of features (e.g., certain substructures) are most indicative of molecular properties. On the other hand, the performance decreases of deep models are less significant, and most deep models can beat tree models after the transformations. We explain this observation as follows. Without the loss of generality, we assume that a linear layer of deep models can map the original molecular feature $x_i$ to the label $y_i$,

$$y_i = W^\top x_i + b, \tag{3}$$

where $W$ and $b$ denote the parameter matrix and the bias term of the linear layer, respectively. And then, we aim to learn a new linear layer mapping the transformed model feature $\widehat{x_i}$ to label $y_i$,

$$y_i = \widehat{W}^\top \widehat{x_i} + b = \widehat{W}^\top \mathcal{Q}x_i + \hat{b}, \tag{4}$$

where $\widehat{W}$ and $\hat{b}$ denote the parameter matrix and the bias term of the new linear layer, respectively. Apparently, to achieve the same results as the original feature, we only have to learn $\widehat{W}$ so that $\widehat{W} = \mathcal{Q}W$ because $\mathcal{Q}^{-1} = \mathcal{Q}^\top$ as an orthogonal matrix, and also $\hat{b} = b$. Therefore, applying the orthogonal transformation to molecular features barely impacts the performance of deep models. The empirical results in Figure 4 confirm this point although some performance changes are observable due to uncontrollable random factors. This explanation inspires us not to mix the molecular features before feeding them into models.

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

## A. The performance of various models on the orthogonally transformed dataset

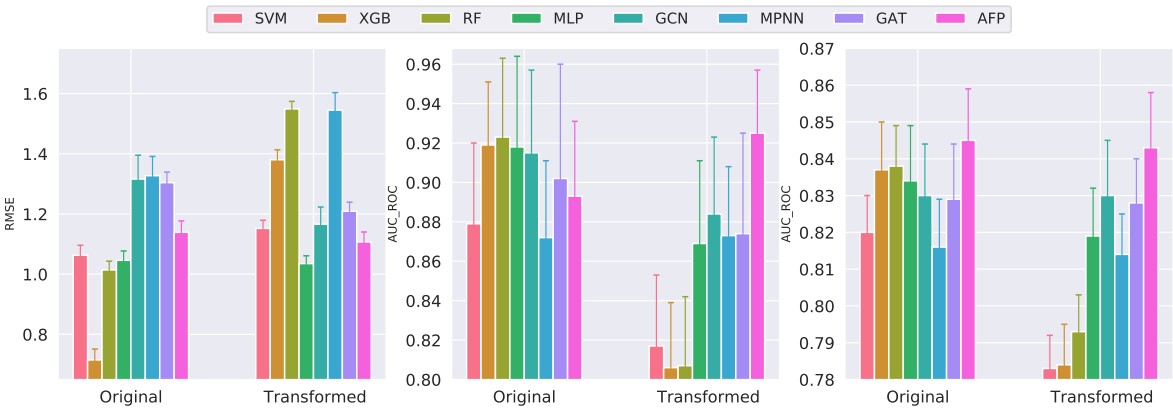

*Figure 4.* The performance of various models on the orthogonally transformed datasets. **Left**: FreeSolv (Regression); **Middle**: ClinTox (Classification); **Right**: Tox21 (Classification). Kindly note that we did not evaluate CNN, RNN, and TRSF on the transformed datasets because we cannot apply the orthogonal transformations to the input SMILES strings.

## B. Experimental Setups

**Fingerprints ⟼ SVM, XGB, RF, and MLP.** Following the common practice (Tian et al., 2022; Pattanaik & Coley, 2020), we feed the concatenation of various molecular fingerprints including 881 PubChem fingerprints (PubchemFP), 307 substructure fingerprints (SubFP), and 206 MOE 1-D and 2-D descriptors (Yap, 2011) to SVM, XGB, RF, and MLP models to comprehensively represent molecular structures, with some pre-processing procedures to remove features (1) with missing values; (2) with extremely low variance (variance < 0.05); (3) have a high correlation (Pearson correlation coefficient > 0.95) with another feature. The retained features are normalized to the mean value of 0 and variance of 1. Additionally, considering that traditional machine models (SVM, RF, XGB) cannot be directly applied in the multi-task molecular datasets, we split the multi-task dataset into multiple single-task datasets and use each of them to train the models. Finally, we report the average performance of these single tasks.

**SMILES strings ⟼ CNN, RNN, and TRSF.** We adopt the 1D CNNs from a recent study (Kimber et al., 2021), which include a single 1D convolutional layer with a step size equal to 1, followed by a fully connected layer. As for the RNN, we use a 3-layer bidirectional gated recurrent units (GRUs) (Cho et al., 2014) with 256 hidden vector dimensions. Additionally, we use the pre-trained SMILES transformer (Honda et al., 2019) with 4 basic blocks and each block has 4-head attentions with 256 embedding dimensions and 2 linear layers. The SMILES are split into symbols (e.g., 'Br', 'C', '=', '(','2') and then fed into the transformer together with the positional encoding (Vaswani et al., 2017).

**2D Graphs ⟼ GCN, MPNN, GAT, and AFP.** As in previous studies (Xiong et al., 2019), we exhaustively utilized all readily available atom/bond features in our 2D graph-based descriptors. Specifically, we have incorporated 9 atom features, including atom symbol, degree, and formal charge, using a one-hot encoding scheme. In addition, we included 4 bond features, such as type, conjugation, ring, and stereo. The resulting encoded graphs were then fed into GCN, MPNN, GAT, and AFP models. Further details on the graph descriptors used in our experiments can be found in (Xiong et al., 2019).

**3D Graphs ⟼ SPN.** We employ the recently proposed SphereNet (Liu et al., 2022) for molecules with 3D geometry. Specifically, for quantum mechanics datasets (QM7 and QM8) that contain 3D atomic coordinates calculated with ab initio Density Functional Theory (DFT), we feed them into SphereNet directly. For other datasets without labeled conformations, we used RDKit (Landrum, 2013)-generated conformations to satisfy the request of SphereNet.

**Datasets splits, evaluation protocols and metrics, hyper-parameters tuning.** Firstly, we randomly split the training, validation, and test sets at a ratio of 8:1:1. And then, we tune the hyper-parameters based on the performance of the validation set. Specifically, we select the optimal hyper-parameters set using the Tree of Parzen Estimators (TPE) algorithm (Ozaki et al., 2020) in 50 evaluations. Due to the heavy computational overhead, GNNs-based models on the HIV and MUV datasets are in 30 evaluations; all the models on the QM7 and QM8 are in 10 evaluations. And then, we conduct 50 independent

runs with different random seeds for dataset splitting to obtain more reliable results, using the optimal hyper-parameters determined before. Similarly, GNNs-based models on the HIV and MUV datasets are in 30 evaluations; all the models on the QM7 and QM8 are in 10 evaluations. Following MoleculeNet benchmark (Wu et al., 2018), we evaluate the classification tasks using the area under the receiver operating characteristic curve (AUC-ROC), except the area under the precision curve (AUC-PRC) on MUV dataset due to its extreme biased data distribution. The performance on the regression task are reported using root mean square error (RMSE) or mean absolute error (MAE). kindly note that we report the average performance across multi-tasks on some datasets because they contain more than one task. Additionally, to avoid the overfitting issue, all the deep models are trained with an early stopping scheme if no validation performance improvement is observed in successive 50 epochs. We set the maximal epoch as 300 and the batch-size as 128.

## C. Related Work

In this section, we elaborate on various molecular descriptors and their respective learning models.

### C.1. Fingerprints-based Molecular Descriptors

Molecular fingerprints (FPs) serve as one of the most important descriptors for molecules. Typical examples include Extended-Connectivity Fingerprints (ECFP) (Rogers & Hahn, 2010) and PubChemFP (Wang et al., 2017). These fingerprints encode the neighboring environments of heavy atoms in a molecule into a fixed bit string with a hash function, where each bit indicates whether a certain substructure is present in the molecule. Traditional models (e.g., tree or SVM-based models) and MLPs can take these fingerprints as 'raw' input. However, the high-dimensional and sparse nature of FPs introduces additional efforts for feature selection when they are fed into certain models. Additionally, it is difficult to interpret the relationship between properties and structures because the hash functions are non-invertible.

### C.2. Linear Notation-based Molecular Descriptors

Another option for molecules is linear notations, among which SMILES (Weininger et al., 1989) is the most frequently-used one owing to its versatility and interpretability. In SMILES, each atom is represented as a respective ASCII symbol; Chemical bonds, branching, and stereochemistry are denoted by specific symbols. However, a significant fraction of SMILES strings does not correspond to chemically valid molecules. As a remedy, a new language named SELF-referencIng Embedded Strings (SELFIES) for molecules was introduced in 2020 (Krenn et al., 2020). Every SELFIES string corresponds to a valid molecule, and SELFIES can represent every molecule. Naturally, RNNs, 1D CNN, and Transformers are powerful deep models for processing such sequences (Wang et al., 2019; Zheng et al., 2019; Honda et al., 2019; Ross et al., 2022; Yüksel et al., 2023). However, the poor scalability of the sequential notations and the loss of spatial information limit the performances of these approaches.

### C.3. 2D and 3D Graph-based Molecular Descriptors

Molecules can be represented with graphs naturally, with nodes as atoms and edges as chemical bonds. Initially, (Duvenaud et al., 2015) first adopted convolutional layers to encode molecular graphs to neural fingerprints. Following this work, (Coley et al., 2017) employs the atom-based message-passing scheme to learn expressive molecular graph representations. To complement the atom's information, (Kearnes et al., 2016) utilized both the atom's and bonds' attributes, and MPNN (Gilmer et al., 2017) generalized it to a unified framework. Also, multiple variants of the MPNN framework are developed to avoid unnecessary loops (DMPNN (Yang et al., 2019)), to strengthen the message interactions between nodes and edges (CMPNN (Song et al., 2020)), to capture the complex inherent quantum interactions of molecules (MGCN (Lu et al., 2019)), or take the longer-range dependencies (Attentive FP (Xiong et al., 2019)). More recently, some hybrid architectures (Rong et al., 2020; Ying et al., 2021; Min et al., 2022) of GNNs and transformers are emerging to capture the topological structures of molecular graphs. Additionally, given that the available labels for molecules are often expensive or incorrect (Xia et al., 2021; Tan et al., 2021; Xia et al., 2022a), the emerging self-supervised pre-training strategies (You et al., 2020; Xia et al., 2022c;b;e; Yue et al., 2022; Liu et al., 2023) on graph-structured data are promising for molecular graph data (Hu et al., 2020; Xia et al., 2023a;b; Gao et al., 2022), just like the overwhelming success of pre-trained language models in natural language processing community (Devlin et al., 2019; Zheng et al., 2022).

The 3D molecular graph is composed of nodes (atoms), and their positions in 3D space and edges (bonds). The advantage of using 3D geometry is that the conformer information is critical to many molecular properties, especially quantum properties.

In addition, it is also possible to directly leverage stereochemistry information such as chirality given the 3D geometries. Recently, multiple works (Schuett et al., 2017; Satorras et al., 2021; Du et al., 2022; Liu et al., 2022; Atz et al., 2021) have developed message-passing mechanisms tailored for 3D geometries, which enable the learned molecular representations to follow certain physical symmetries, such as equivariance to translations and rotations. However, the calculation cost, alignment invariance, uncertainty in conformation generation, and unavailable conformations of target molecules limited the applicability of these models in practice.

## D. Discussion and Conclusion

In this paper, we perform a comprehensive benchmark of representative models on molecular property prediction. Our results reveal that traditional machine learning models, especially tree models, can easily outperform well-designed deep models in most cases. These phenomena can be attributed to the unique patterns of molecular data and different inductive biases of various models. Specifically, the target function mapping molecules to properties are non-smooth, and some small changes can incur significant property variance. Deep models struggle to learn such patterns. Additionally, molecular features carry meanings individually and deep models would undesirably mix different dimensions of molecular features. Our study leaves an open question for future research: Can our findings and methods be generalized to other AIDD tasks including drug-target interactions (DTIs) prediction (Ozturk et al., 2018; Xia et al., 2022d), drug-drug interactions (DDIs) prediction (Li et al., 2021), and protein representation learning (Hu et al., 2022; Tan et al., 2023)?

