# OpenReview forum: "Why Deep Models Often Cannot Beat Non-deep Counterparts on Molecular Property Prediction?"
_ICML.cc/2023/Workshop/IMLH — IMLH 2023 PosterShortPaper_

### Official Review · Reviewer_McD9 · 2023-06-05
**Why Deep Models Often Cannot Beat Non-deep Counterparts on Molecular Property Prediction?**

**Rating:** 9
**Confidence:** 4

**Review:**

In this paper, the authors benchmark 12 models on 14 molecule datasets to compare deep models and non-deep models for the molecular property prediction task. Through the comprehensive comparison and in-depth analysis, the three key observations are obtained: 1) non-deep models generally outperform deep models; 2) the irregular molecule data pattern is an important failure reason; 3) tree models using molecular fingerprints as inputs tend to perform better than other competitors. Overall, the motivation of this paper is clear. The observations are valuable for the practice of the molecular property prediction task.

---

### Meta-Review · Area_Chair_Wh7s · 2023-06-20

**Recommendation:** Accept (Poster)
**Confidence:** 5

**Metareview:**

This paper presents a comprehensive benchmark study for molecular property prediction, highlights three key observations, and is expected to provide valuable insights for the field.

---

### Decision · Program_Chairs · 2023-06-20

Accept (Poster Short Paper)